# Revealing Changes in Ovarian and Hemolymphatic Metabolites Using Widely Targeted Metabolomics between Newly Emerged and Laying Queens of Honeybee (*Apis mellifera*)

**DOI:** 10.3390/insects15040263

**Published:** 2024-04-11

**Authors:** Shiqing Zhong, Luxia Pan, Zilong Wang, Zhijiang Zeng

**Affiliations:** 1Honeybee Research Institute, Jiangxi Agricultural University, Nanchang 330045, China; shiqing_zhong@163.com (S.Z.); luxia_pan@163.com (L.P.); wzlcqbb@126.com (Z.W.); 2Jiangxi Province Key Laboratory of Honeybee Biology and Beekeeping, Nanchang 330045, China

**Keywords:** queen bee, ovary, hemolymph, metabolomics

## Abstract

**Simple Summary:**

The queen bee specializes in colony reproduction and undergoes behavioral and physical changes following egg-laying. Metabolomics is a high-throughput technique that can unveil the intricate connection between biological phenotypic states and an organism’s small molecules. We are intrigued to determine the characteristic substances present in the hemolymph and ovary, as well as to understand how their rapid metabolism contributes to the process of egg-laying by queens. In this study, we reared *Apis mellifera* queens from three distinct time periods: newly emerged queen (NEQ), newly laying queen (NLQ), and old laying queen (OLQ). Using widely targeted metabolomics, we found that NLQ and OLQ exhibited significant metabolite alterations compared to NEQ, such as up-regulated expression of carnitines, fatty acids, and some antioxidants, and significantly down-regulated expression of amino acids, etc. The results of this study provide a novel perspective for analyzing the oviposition mechanism of queens.

**Abstract:**

The queen bee is a central and pivotal figure within the colony, serving as the sole fertile female responsible for its reproduction. The queen possesses an open circulatory system, with her ovaries immersed in hemolymph. A continuous and intricate transportation and interchange of substances exist between the ovaries and hemolymph of queen bees. To determine the characteristic metabolites in the hemolymph and ovary, as well as understand how their rapid metabolism contributes to the process of egg-laying by queens, we reared *Apis mellifera* queens from three different age groups: newly emerged queen (NEQ), newly laying queen (NLQ), and old laying queen (OLQ). Using widely targeted metabolomics, our study revealed that the laying queen (NLQ and OLQ) exhibited faster fatty acid metabolism, up-regulated expression of antioxidants, and significant depletion of amino acids compared to the NEQ. This study revealed that the levels of carnitine and antioxidants (GSH, 2-O-α-D-glucopyranosyl-L-ascorbic acid, L-ascorbic acid 2-phosphate, etc.) in the NLQ and OLQ were significantly higher compared to NEQ. However, most of the differentially expressed amino acids, such as L-tryptophan, L-tyrosine, L-aspartic acid, etc., detected in NLQ and OLQ were down-regulated compared to the NEQ. Following egg-laying, pathways in the queens change significantly, e.g., Tryptophan metabolism, Tyrosine metabolism, cAMP signaling pathway, etc. Our results suggest that carnitine and antioxidants work together to maintain the redox balance of the queen. Additionally, various amino acids are responsible for maintaining the queen’s egg production.

## 1. Introduction

The colony’s reproduction and growth depend on the queen, who must maintain high rates of egg production. *Apis mellifera* queens have the capacity to lay 1500–2000 eggs daily [1,2,3]. The queens’ ovaries have a rapid metabolism, allowing them to lay more eggs in 24 h than their own body weight. However, in the practice of beekeeping, the queen’s egg-laying ability is influenced by various factors, including rearing methods [4,5], rearing season [6], and comb age [7,8], with the primary factor being the queens themselves, their genome [9,10], age [11], health [11,12], etc.

The queen bee possesses a highly developed reproductive system. The fully developed ovaries of the queen consist of a pair of large, pear-shaped ovarioles. The estimated number of ovarioles per ovary in *Apis mellifera* (*A. mellifera*) is generally between 100 and 180 [1,13,14]. The ovary is essential for the queen; the weight of the ovaries and the number of ovarioles are common indicators of the reproductive capacity of honeybee queens [15,16]. Additionally, the developmental and reproductive performance of queens is usually measured by the use of genes highly expressed in the ovaries, such as *Hexamerin* (*Hex*) *110* [17,18,19], *Hex 70a* [20,21], and *Vitellogenin* (*Vg*) [22,23].

The honeybee has an open circulatory system with a fluid similar to blood, called hemolymph, and its ovary is immersed in it [24,25]. The hemolymph is essential to honeybee immunity and is widely used as a research medium for investigating viral, bacterial, and parasitic infections in honeybees [26,27,28,29,30]. It is also used to transfer micro- and macro-elements and has been employed in the assessment of metal pollution [31]. The main components of honeybee hemolymph are proteins, lipids, carbohydrates, and amino acids, as well as water [32]. Therefore, the hemolymph facilitates the transportation of various nutrients to the queen’s ovaries. For example, vitellogenin (Vg), synthesized by the honeybee’s fat body, plays a role in oocyte growth and embryo energy supply [33,34] and is taken up by oocytes in the honeybee ovary [35,36].

Metabolomics is a high-throughput technique that can unveil the intricate connection between biological phenotypic states and an organism’s small molecules. In honeybee metabolomics research, commonly employed experimental samples include the brain, head part, gut, and hemolymph [37]. These studies have primarily focused on investigating the impacts of insecticide and herbicide exposure [38,39,40,41], infections [42,43,44], dietary changes [45,46], and other factors on honeybees. Previous studies have also analyzed the metabolic profiles of the spermatheca [47] and ovaries [48] in the laying queen (LQ). However, a lack of research exists into the metabolite alterations in ovaries and hemolymph between newly emerged queen (NEQ) and LQ (*A. mellifera*).

There is a continuous and intricate transportation and interchange of substances between the ovaries and hemolymph of queens, which is intimately linked with the process of oviposition. Therefore, we are intrigued to determine the characteristic substances present in the hemolymph and ovary, as well as to understand how their rapid metabolism contributes to the process of egg-laying by queens. In this study, we reared *A. mellifera* queens from three distinct time periods. We aimed to identify differentially expressed metabolites (DEMs) and important pathways between NEQ and LQ through metabolomic analysis. This provides a novel perspective for analyzing the oviposition mechanism of queens.

## 2. Materials and Methods

### 2.1. Acquisition of Experimental Samples

The honeybees (*A. mellifera ligustica*) were maintained at the Honeybee Research Institute of Jiangxi Agricultural University (28.46° N, 115.49° E). The queens were collected from three different age groups: newly emerged queen (NEQ, zero days old), newly laying queen (NLQ, approximately one month old), and old laying queen (OLQ, approximately one year old). The NEQ was obtained through the standard artificial queen rearing method [2]. Each age group had six replicates. The NLQ and OLQ were natural mating. Six NEQs, three NLQs, or three OLQs were chosen for each ovary sample. Six queens were selected for every hemolymph sample.

### 2.2. Gathering the Ovary and Hemolymph of the Queen Bee

The queens were attached to a beeswax dish with insect pins after the wings and legs were removed. Subsequently, their ovaries were then promptly preserved in liquid nitrogen with freezing tubes after being stripped using sterile scissors and tweezers. For hemolymph, after comparing the various methods [49,50,51], we chose the most effective method, i.e., dorsal sinus with a capillary hemolymph sampling (DCHS). Firstly, the queens were anesthetized at −20 °C for 10 min. Then, the connecting membrane between the third and fourth dorsal segments was pierced using a capillary. The hemolymph was introduced into the capillary and subsequently extruded into a freezing tube before being stored in liquid nitrogen. All processes are carried out on an ultra-clean table.

### 2.3. Widely Targeted Metabolomics Profiling Methods and Conditions

T3 UPLC Conditions: The LC-ESI-MS/MS system (UPLC, ExionLC AD, https://sciex.com.cn/, accessed on 17 July 2022; MS, QTRAP^®^ System, https://sciex.com/, accessed on 17 July 2022) was utilized for the analysis of sample extracts. The experimental parameters were set as follows: UPLC included a Waters ACQUITY UPLC HSS T3 C18 column (1.8 μm, 2.1 mm × 100 mm) maintained at a temperature of 40 °C with a flow rate of 0.4 mL/min. The sample was injected either at a volume of 2 μL or 5 μL using a solvent system consisting of water with 0.1% formic acid and acetonitrile with 0.1% formic acid. The gradient program included the following steps: 95:5 *v*/*v* at 0 min, 10:90 *v*/*v* at 10.0 min, 10:90 *v*/*v* at 11.0 min, 95:5 *v*/*v* at 11.1 min, 95:5 *v*/*v* at 14.0 min.

Considering it could acquire MS/MS spectra in an information-dependent manner (IDA) during an LC/MS experiment, the Triple TOF mass spectrometer was employed. In this mode, the MS/MS spectra acquisition is triggered by certain conditions, and the acquisition software (TripleTOF 6600, AB SCIEX) continually assesses the full scan survey MS data as it is being collected. In this mode, the acquisition software (TripleTOF 6600, AB SCIEX) continuously evaluates the full scan survey MS data while collecting it and triggers the acquisition of MS/MS spectra based on preselected criteria. Twelve precursor ions with intensities higher than 100 were chosen for fragmentation at a collision energy (CE) of 30 V during each cycle. This produced 12 MS/MS events with product ion accumulation times of 50 msec each. The following parameters were configured for the ESI source: Ion source gas 1 at 50 Psi, Ion source gas 2 at 50 Psi, Curtain gas at 25 Psi, source temperature at 500 °C, and Ion Spray Voltage Floating (ISVF) set to either 5500 V or −4500 V in positive or negative modes, respectively.

The QTRAP^®^ LC-MS/MS System, functioning as a triple quadrupole-linear ion trap mass spectrometer with an ESI Turbo Ion-Spray interface, was used to acquire LIT and triple quadrupole (QQQ) scans. Under the software control of Analyst 1.6.3 (Sciex), the apparatus functioned in both positive and negative ion modes. The 500 °C source temperature, 5500 V (positive) and −4500 V (negative) ion spray voltage (IS), 50, 50, and 25.0 psi for ion source gas I (GSI), gas II (GSII), and curtain gas (CUR), respectively, and high collision gas (CAD) were the settings for the ESI source. In QQQ and LIT modes, 10 and 100 μmol/L polypropylene glycol solutions were used for instrument tuning and mass calibration, respectively. Specific MRM transitions were monitored for each period based on the eluted metabolites.

### 2.4. Data Analysis and Statistics

DEMs were identified based on criteria, including |Log_2_FC| > 1.0 and VIP ≥ 1, with VIP values obtained from the OPLS-DA results utilizing the R package MetaboAnalystR. Analyzing data and plotting ring and PCA plots using the Metware Cloud platform (https://cloud.metware.cn/, accessed on 21 December 2023).The Kyoto Encyclopedia of Genes and Genomes (KEGG) Compound database (http://www.kegg.jp/kegg/compound/, accessed on 7 January 2024) was used to annotate the identified metabolites. Pathways containing significantly regulated metabolites were subjected to metabolite set enrichment analysis (MSEA), with significance assessed using the *p*-values derived from the hypergeometric test.

## 3. Results

### 3.1. Identification of Metabolites and Widely Targeted Metabolomic Analysis

The widely targeted metabolomics analysis of the 36 samples from the ovary and hemolymph revealed a total of 3312 metabolites. These included 24 different classes of metabolites (Figure 1A), such as “Amino acid and its metabolites”, “Benzene and substituted derivatives”, “Heterocyclic compounds”, “Organic acid and its derivatives”, “Aldehyde, Ketones, and Esters”, etc. After performing principal component analysis (PCA) on ovary and hemolymph, we found that there was a clear difference between the metabolites of LQ and NEQ, but the difference between NLQ and OLQ is relatively small (Figure 1B). We conducted an orthogonal partial least squares discriminant analysis (OPLS-DA) concurrently (Appendix A). These findings imply that the OPLS-DA models are dependable (Q2 ≥ 0.9810) and can be utilized for additional research on differentially expressed metabolites (DEMs).

### 3.2. DEMs in Ovary and Hemolymph

The number of DEMs in the *A. mellifera* ovary (OV) in the NLQ vs. NEQ and OLQ vs. NEQ groups was 1202 and 1209, respectively. In the *A. mellifera* hemolymph (HE), the number of DEMs in the NLQ vs. NEQ and OLQ vs. NEQ groups was 803 and 817, respectively (Figure 2A). To identify the common key metabolites in NLQ vs. NEQ and OLQ vs. NEQ, we utilized Venn diagrams to classify the up-regulated and down-regulated metabolites. In comparison to NEQ, a total of 248 metabolites were found to be up-regulated in OV from NLQ and OLQ (Figure 2B, OV_Up). However, a total of 817 metabolites were down-regulated in OV from NLQ and OLQ compared to NEQ (Figure 2B, OV_Down). Furthermore, the results showed an upregulation of 144 metabolites in HE from NLQ and OLQ when compared to NEQ (Figure 2B, HE_Up). In addition, 447 down-regulated metabolites were found in HE from NLQ and OLQ compared to NEQ (Figure 2B, HE_Down).

### 3.3. Analysis of Key Metabolites with Consistent Trends

To better understand the trends in the expression of honeybee queen ovary and hemolymph metabolites, we selected some of the critical DEMs to create heatmaps. In the ovaries of NLQ and OLQ, the expression of the fatty acyls (FA) and glycerophospholipid (GP) classes was much higher than NEQ (Appendix A), including stearoyl-L-carnitine, carnitine C2:0, carnitine C3:0, FFA (12:1), PC (8:0/8:0), PC (16:0/2:0), LPC (0:0/18:2), LPE (16:1/0:0), LPI (14:0/0:0), etc. Moreover, the levels of glutathione reduced form, glutathione oxidized, oxiglutatione, and L-ascorbic acid 2-phosphate were significantly increased in the ovaries of LQ compared to NEQ (Figure 3A). The metabolites highly expressed in the NEQ ovary are mainly concentrated in amino acid classes. In particular, DEMs such as N1-acetyl-5-methoxykynuramine (AMK), 6-hydroxymelatonin, 2-hydroxymelatonin, (+)-fluprostenol, and testosterone glucuronide were exclusively detected in the ovaries of NEQ (Figure 3B).

In hemolymph, the up-regulated metabolites were primarily FA and GP (Appendix A), in addition to the usual nucleotides, carbohydrates, and their metabolites. The metabolites highly expressed in the NEQ hemolymph are also mainly concentrated in amino acid classes. Furthermore, dopamine, L-dopa, DL-2-methylglutamic acid, and norepinephrine were only found in the hemolymph of NEQ (Figure 3D).

### 3.4. Important Metabolic Pathways Affected by Metabolite Variations

We employed two sets of intersecting metabolites in the Venn diagram, NLQ vs. NEQ and OLQ vs. NEQ (Appendix A), to perform KEGG analysis. The results demonstrated that, following egg-laying, pathways in the ovary and hemolymph of queens change significantly (*p* < 0.05), such as Tryptophan metabolism, Tyrosine metabolism, cAMP signaling pathway, Thyroid hormone synthesis, Phosphonate and phosphinate metabolism, Pantothenate and CoA biosynthesis, and Aminoacyl-tRNA biosynthesis (Figure 4).

## 4. Discussion

In this study, it was observed that the levels of carnitine in the ovaries of LQ were significantly higher compared to NEQ. The trend of free fatty acids, such as FFA (18:2), FFA (18:4), FFA (12:1), etc., in queen bee ovaries was consistent with carnitine. Carnitine acts as a carrier molecule for activated fatty acids, promoting fatty acid oxidation within the mitochondria [52,53]. L-carnitine has been extensively studied compared to other carnitines, with numerous investigations demonstrating its efficacy in increasing oocyte quality and expediting blastocyst formation during in vitro oocyte maturation across various species, including pigs [54,55], cattle [56,57], mice [58,59], etc. Notably, Li et al. conducted single-cell metabolomic sequencing on mouse oocytes at three different developmental stages and confirmed the heightened levels of carnitines during oocyte meiosis [60]. Furthermore, DL-carnitine has been identified as a stimulant for oviposition in *Drosophila melanogaster* (*D. melanogaster*) by Geer and Dolph [61]. Therefore, it is postulated that carnitine plays a crucial role in promoting fatty acid metabolism during queen oviposition, ensuring a continuous supply of fatty acids for rapid oocyte development. Additionally, owing to its antioxidant properties [62,63], carnitine may protect against oxidative stress within the ovary.

Oxidative stress occurs when the body produces an excessive amount of reactive oxygen species (ROS) or fails to effectively eliminate the excess ROS. Antioxidants can regulate ROS levels and maintain redox balance in the body. In mammals, antioxidants are regarded as pivotal elements in ovarian physiological metabolism. Enzymatic antioxidants, including catalase (CAT) [64,65], superoxide dismutase (SOD) [66,67], glutathione-S-transferase (GST) [68,69], etc., have been demonstrated to contribute significantly to the development of oocytes. Glutathione (GSH) [70,71], ascorbic acid (AA) [72,73], melatonin (MLT) [74,75,76], etc., which are non-enzymatic antioxidants, safeguard oocytes against damage caused by ROS and enhance the quality of oocytes. In honeybees, the activities of the queen ovary CAT, SOD, and GST transcripts increase with the reproductive maturity of the queen [77]. Weirich et al. conducted assays and found no significant differences in the activities of CAT, SOD, and GST in the hemolymph between NEQ and LQ [78]. Non-enzymatic antioxidants in honeybees have received limited attention, despite their significance. Notably, GSH has been exclusively identified in the ovaries of LQ. Consequently, we propose a hypothesis that LQ produces substantial amounts of ROS and necessitates elevated levels of GSH to maintain homeostasis. Honeybees have the ability to produce AA [79]. Supplementation of honeybee colonies with AA during the winter and early spring has been shown to enhance honeybees’ resistance to oxidative stress, resulting in a reduced infestation rate by *Varroa destructor* [80,81]. We only detected L-AA in the hemolymph of queens, and there was a significant difference between NLQ vs. NEQ groups, but not between OLQ vs. NEQ groups. 2-O-α-D-glucopyranosyl-L-ascorbic acid (AA-2G), the stable AA derivative, exhibits increased levels in LQ hemolymph compared to NEQ. AA-2G possesses inherent resistance against oxidative stress and can also be enzymatically converted into the highly potent antioxidant AA [82,83]. In the LQ ovary, we found another AA derivative, L-ascorbic acid 2-phosphate (AA-2P). AA-2P is highly stable under normal cell culture conditions and exhibits a longer duration of vitamin C activity compared to AA, thereby effectively preventing oxidative DNA damage [84,85,86]. MLT is mostly found in the head of honeybees. This study revealed the absence of MLT in the ovaries and hemolymph of queens. Interestingly, three distinct MLT metabolites, namely AMK, 6-hydroxymelatonin, and 2-hydroxymelatonin, were exclusively identified within the ovaries of NEQ. This phenomenon is likely to be attributed to the LQ’s uninterrupted oviposition and their absence of diel rhythmicity, while the NEQ demonstrates free-running circadian rhythms [87]. MLT can enhance honeybees’ ability to resist cold tolerance [88] and imidacloprid [89]. Further study is needed to determine whether MLT also affects egg production in queens, similar to its effects on *D. melanogaster* [90].

The antioxidant capacity is closely associated with the queen’s high egg-laying performance, while the amino acid requirement also reflects the speed of the queen’s high egg-laying. Amino acids play a crucial role in facilitating ovarian functions through their contribution to protein synthesis, hormone production, immune regulation, and other essential processes [91,92,93,94,95]. Honeybees, similar to other animals, have 10 essential amino acids (EAAs) that need to be supplemented in order to maintain normal growth, development, and reproduction: lysine (Lys), histidine (His), tryptophan (Try), valine (Val), arginine (Arg), isoleucine (Ile), leucine (Leu), methionine (Met), phenylalanine (Phe), and threonine (Thr) [96,97]. In our study, we found that most of the differentially expressed amino acids detected in LQ were down-regulated compared to NEQ. The EAAs that were down-regulated in hemolymph were L-Lys, L-His, L-Try, L-Val, and L-Ile. The down-regulated amino acids shared in the ovary and hemolymph include L-Try, L-tyrosine (L-Tyr), L-aspartic acid (L-Asp), L-citrulline (L-Cit), L-homocitrulline (L-HC), L-homoarginine (L-HA), and L-norleucine (L-NLE). Hrassnigg et al. also found that free amino acids without proline and some free EAAs in hemolymph were decreased in LQ compared to NEQ, e.g., Tyr, Arg, Ile, Leu, Lys, and His [98]. Egg-laying by queens can require substantial amounts of amino acids, resulting in a down-regulation of most amino acids compared to NEQ. Sang and King discovered that egg production in *D. melanogaster* requires all 10 EAAs and that non-essential amino acids also maintain egg production [99]. Alves et al. showed that a diet deficient in a single amino acid (Arg, Ile, Leu, Lys, Phe, Thr, and Try) resulted in a sharp drop in egg production in *D. melanogaster*, whereas deficiencies in Met, His, and Val declined more gradually in egg production [100]. The queen can acquire EAAs from royal jelly [101,102], enabling her to sustain her egg-laying capacity for an extended period of time.

The metabolism of Try and Tyr shows a great impact after egg-laying in queens. Metabolites in the tryptophan and tyrosine metabolic pathways are predominantly down-regulated in LQ compared to NEQ. This is most likely to be due to the sustained consumption of Try, Tyr, and their metabolites by the queen in her reproductive role. Try is an EAA for all insects, playing crucial roles in various physiological processes, including reproduction. Try can promote yolk formation [103], induce and promote ovarian development [104,105], and regulate queen ovary metabolism [48]. Tyr is a non-EAA that is synthesized via Phe. Tyr is the key precursor for tanning *Aedes aegypti* eggs [106]. Tyr deficiency significantly reduces insect egg production and affects egg hatchability [107,108,109]. Tyr in royal jelly may promote workers’ ovarian development in queenless colonies [110]. It is also likely that queens laying eggs require large amounts of Tyr from royal jelly. L-Lys, L-His, L-Val, L-Ile, L-aspartic acid, etc., may also have a close correlation with queens laying eggs, but additional verification is required.

## 5. Conclusions

The high rate of fatty acid metabolism and the high expression of antioxidants in the ovary and hemolymph provide material security and stress resistance for laying queens. Various amino acids are responsible for maintaining the queen’s egg production; oviposition significantly depletes amino acid reserves in queens. Following egg-laying, tryptophan and tyrosine metabolic pathways are significantly affected in queens’ ovary and hemolymph, underscoring the pivotal role of tryptophan and tyrosine in facilitating the queen bee’s reproductive capacity.

## Figures and Tables

**Figure 1 insects-15-00263-f001:**
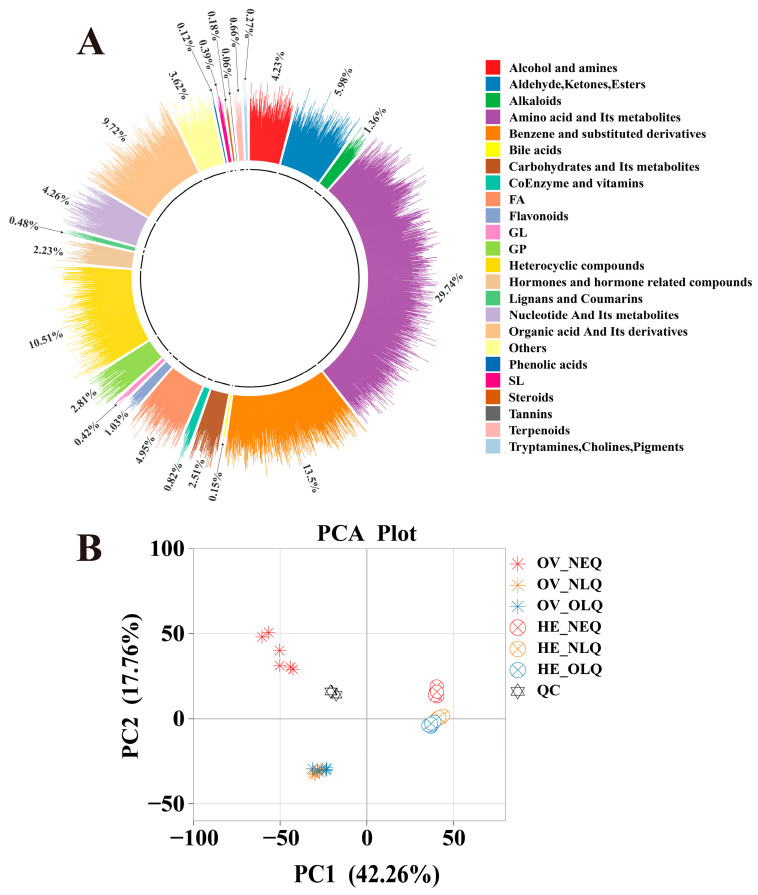
Ring plot and PCA plots in all groups. (**A**) Ring plot of metabolites in queens’ ovary and hemolymph. (**B**) PCA plots in all groups of queens’ ovary and hemolymph. “OV”: Ovary; “HE”: Hemolymph.

**Figure 2 insects-15-00263-f002:**
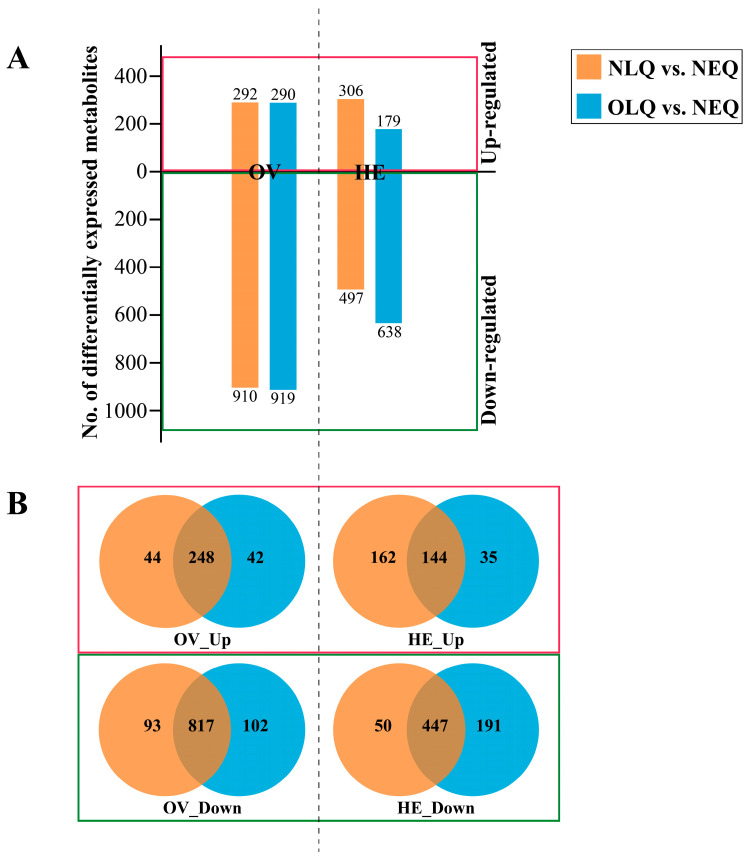
Analysis of DEMs. (**A**) Number of DEMs in all groups. (**B**) The Venn diagram shows the common DEMs between NLQ vs. NEQ and OLQ vs. NEQ. “OV_Up”: The metabolites that expressed higher in the OV of NLQ and OLQ than in the OV of NEQ; “HE_Up”: The metabolites that expressed higher in the HE of NLQ and OLQ than in the MH of NEQ; “OV_Down” and “HE_Down”: The down-regulated metabolites that showed a lower expression level in the OV or HE of NLQ and OLQ compared to those in NEQ.

**Figure 3 insects-15-00263-f003:**
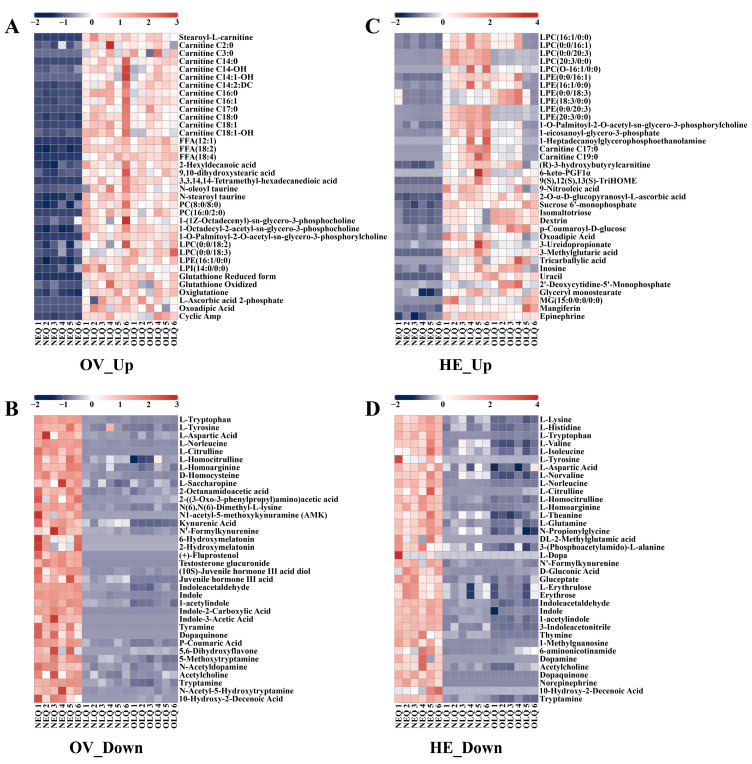
The heatmaps show the representatively metabolites in the ovary and hemolymph. (**A**) The DEMs that showed higher expression levels in the ovary of NLQ and OLQ compared to that of NEQ. (**B**) The DEMs that showed lower expression levels in the ovary of NLQ and OLQ compared to that of NEQ. (**C**) The DEMs that showed higher expression levels in the hemolymph of NLQ and OLQ compared to that of NEQ. (**D**) The DEMs that showed lower expression levels in the hemolymph of NLQ and OLQ compared to that of NEQ.

**Figure 4 insects-15-00263-f004:**
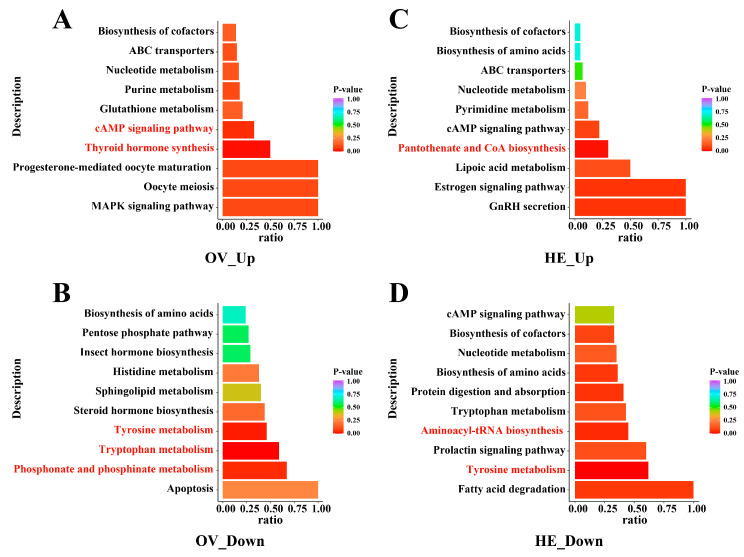
The bar plot of KEGG metabolic pathways commonly shared by NLQ vs. NEQ and OLQ vs. NEQ. (**A**) Up-regulated metabolic pathways in the ovary. (**B**) Down-regulated metabolic pathways in the ovary. (**C**) Up-regulated metabolic pathways in the hemolymph. (**D**) Down-regulated metabolic pathways in the hemolymph. The redder colors of the bar indicate pathways that are more important. Pathways marked in red font indicate significant differences. *p* < 0.05 was considered statistically significant.

## Data Availability

The data presented in this study are available in this article.

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
