# Peer review of "Revealing Changes in Ovarian and Hemolymphatic Metabolites Using Widely Targeted Metabolomics between Newly Emerged and Laying Queens of Honeybee (*Apis mellifera*)"

_insects, 2024, doi:10.3390/insects15040263_

Round 1
Reviewer 1 Report
Comments and Suggestions for Authors
Zhong et al.'s study on metabolomic differences between newly emerged and laying honey bee queens is a very well done study and was a pleasure to review. Other than requesting the authors to carefully lookover the abbreviations in the manuscript, I have nothing of note or concern about this research. The manuscript is really well written and aptly discussed.
Reviewer 2 Report
Comments and Suggestions for Authors
In this study, the authors present some important findings. However, some comments should be addressed to improve the quality of this article.
Title:
- Modefy to Revealing changes in ovarian and hemolymphatic metabolites using widely targeted metabolomics between newly emerged and laying queens of honey bee (Apis mellifera)
- Please use one form (honey bee or honeybee) through the manuscript.
Abstract:
- Should include background, objective, materials and methods, results, and conclusion. Please include the missed.
Introduction:
- Line 38-39: Delete this sentence.
- Line 41: Delete (A. mellifera).
- Line 45: The author can use the following references:
- Taha, E.-K.A. Studies on honey bee (Apis mellifera L.). Ph.D. Thesis, Faculty of Agriculture, Tanta University, Kafrelsheikh, Egypt, 2005, 151pp.
- Taha, E-K.A., Shawer, M.B., Taha, R., Elashmawy, A., Gaber, S., Mousa, K. Comb age significantly influences the emergency queen rearing, morphometric and reproductive characteristics of the queens. Journal of Apicultural Research, 2024, 10.1080/00218839.2024.2336376
Materials and Methods
- Line 77: The subspecies of honey bee should mentioned.
- Line 82: Six NEQs, three NLQs, or three OLQs were chosen for each ovary sample. Six queens were selected for every hemolymph sample. Aren't they very low numbers????
- The issue of methods is the very low number of replicates.
- I can't find the statistical analysis section.
Results
- The statistical analysis does not appear in Figs.
Discussion
- Discussion needs some improvements.
- Line 289: Aedes aegypti write it in italic form.
Moderate editing of English language required
Reviewer 3 Report
Comments and Suggestions for Authors
The authors present a study on metabolites of three different groups of honey bee queens, newly emerged ones, newly laying and old laying. It sheds new light on the metabolism of honey bee queens. Understanding honey bee queen metabolism is important as they are the only fertile females within the colony.
Some specific comments:
Abstract:
L24-26: Please rephrase. The sentence starts with “to” but does not end.
Introduction:
L48-56: It is also used to transfer micro– and macro- elements. And was also used to asses metal pollution. Comp. Biochem. Physiol. C Toxicol. Pharmacol. 239; 108852 https://doi.org/ 10.1016/j.cbpc.2020.108852.
L70: I am unsure if “reared” is the correct term here. It also implies to the abstract.
Materials and Methods:
L78-81: Why were zero-day old queens chosen? It would be much better to have chosen queens that were at least five days old. It could be that some metabolism changes happen between the time of emergence and the time that the queens mate.
L90: Should it say -20?
Results:
L154-157: NLO is probably supposed to be NLQ.
L155-158: It should be very clearly stated whether you are talking about hemolymph or ovaries.
L193: Where is Table S2?
L192-197: In the text it is stated that the differences are significant at level of p<0.05. However, in Figure 4. P-value is 0.05 only for upregulated metabolites in ovary?
L205-2019: You have used a zero-day old queens and mated ones. What happens with carnitine in unmated queens that are a month old? Does the metabolomic profile also changes? Could this be a consequence of mating or maturation? This can be applied to all metabolites that showed significant differences between NEQ on one side and NLQ and OLQ on the other, while at the same time there were no differences between NLQ and OLQ (AA)
L237: Please define abbreviations prior to their use. In this case AA (ascorbic acid)
L264-271: When you describe down-regulated you actually mean that you found less of these amino acids in LQ compared to NEQ? Could it be that these amino acids are higher in NEQ as they are required for queens’ maturation?
L272-278: If substantial amounts of EAA are required for laying eggs and the missing a single one can significantly influence the number of laid eggs, wouldn’t it be expected that the amount of EAA is higher in LQ compared to NEQ? It could also be that they use a lot of them and that is the reason for their lower presence in LQ?
L278: How can EAA have a toxic effect on workers? Do you maybe mean that the lack of EAA can have toxic effects?
L282-291: Could the explanation for Tyr apply to all EAA mentioned in L272?
Figures:
Figure 1A. It would be better for the legend to show the compounds in descending order of abundance. At least for me it would be easier to follow.
Figure 1B. Please choose a different sign for ovaries and hemolymph. Then you can have the same color for NEQ, NLQ and OLQ. This would make the figure very easy to understand and follow.
Figure 4. Clearly mark what p-values stand for. If they are for the significance levels then they are contradictory to the ones stated in the text.
Conclusion
Conclusions section should be extended.
Round 2
Reviewer 2 Report
Comments and Suggestions for Authors
The manuscript has been improved
Comments on the Quality of English LanguageThe manuscript has been improved